# Thermo-Fluid-Dynamic Modeling of the Melt Pool during Selective Laser Melting for AZ91D Magnesium Alloy

**DOI:** 10.3390/ma13184157

**Published:** 2020-09-18

**Authors:** Hongyao Shen, Jinwen Yan, Xiaomiao Niu

**Affiliations:** 1The State Key Laboratory of Fluid Power and Mechatronic Systems, College of Mechanical Engineering, Zhejiang University, Hangzhou 310027, China; 21825032@zju.edu.cn (J.Y.); nxm@zju.edu.cn (X.N.); 2Key Laboratory of 3D Printing Process and Equipment of Zhejiang Province, College of Mechanical Engineering, Zhejiang University, Hangzhou 310027, China

**Keywords:** numerical simulation, selective laser melting, AZ91D, melt pool, Marangoni flow

## Abstract

A three dimensional finite element model (FEM) was established to simulate the temperature distribution, flow activity, and deformation of the melt pool of selective laser melting (SLM) AZ91D magnesium alloy powder. The latent heat in phase transition, Marangoni effect, and the movement of laser beam power with a Gaussian energy distribution were taken into account. The influence of the applied linear laser power on temperature distribution, flow field, and the melt-pool dimensions and shape, as well as resultant densification activity, was investigated and is discussed in this paper. Large temperature gradients and high cooling rates were observed during the process. A violent flow occurred in the melt pool, and the divergent flow makes the melt pool wider and longer but shallower. With the increase of laser power, the melt pool’s size increases, but the shape becomes longer and narrower. The width of the melt pool in single-scan experiment is acquired, which is in good agreement with the results predicted by the simulation (with error of 1.49%). This FE model provides an intuitive understanding of the complex physical phenomena that occur during SLM process of AZ91D magnesium alloy. It can help to select the optimal parameters to improve the quality of final parts and reduce the cost of experimental research.

## 1. Introduction

There is a growing interest in the use of magnesium alloys in automotive and aerospace industries, a reason behind which is their very low density compared to aluminum alloys (2/3 of density at room temperature) and steel (2/9 of density at room temperature) [1]. The low density gives them high specific strength and modulus, coupled with their excellent thermal and damping characteristics, making them a potential candidate for low-temperature structural applications in the aerospace and automotive industries [2,3,4]. However, they also have defects such as low creep resistance and easy corrosion, which limit their application in industrial and medical fields. Therefore, alloy elements such as Nd and Zn are added to make them suitable for these applications [5,6]. Magnesium AZ91A, B, C, D, and E have the same nominal compositions but differ in ranges and/or specified impurity limits. This high-purity alloy has excellent corrosion resistance. It is the most commonly used magnesium die-casting alloy. The compositions of impurities in the AZ91D magnesium alloy used in this work are listed in Table 1.

Casting and forming manufacturing are traditional methods for manufacturing magnesium alloys. However, these two methods both suffer from some common disadvantages. Forming provides better mechanical properties than casting, but it suffers from oxidation and induced anisotropy. Casting is prone to defects such as porosity, which reduces the mechanical properties of the manufactured part [7]. This drives the research of new manufacture methods for magnesium alloys, and Addictive Manufacture (AM) is a potential choice.

In the international standard ISO/ASTM 52,900 [8], Additive Manufacturing (AM) is defined as a “process of joining materials to make parts from 3D model data, usually layer upon layer”. It allows building companies to produce geometrically complex structures, to vary materials within a component according to its functions, and to automate the construction process starting from a digital model [9]. Selective laser melting (SLM) is a specific AM technique which utilizes a high-power-density laser to fully melt and fuse metallic powders layer by layer to produce high-freedom-shape parts with near-full density [10,11,12]. With additive support structures and unirradiated powders, SLM technology has the ability to manufacture metal parts of complex shape that would otherwise not be produced by using conventional manufacturing methods [13,14,15,16]. Figure 1 shows a schematic of SLM process.

Lots of experimental researches on magnesium alloy SLM forming have been carried out, most of which are concentrated on the parts, e.g., the compactness of the parts, and the mechanical properties, such as the yield strength of the material [17,18,19,20]. Mahesh et al. studied the effect of preheating and layer thickness on magnesium SLM. When the layer thickness is 0.15–0.20 mm and 0.25–0.30 mm, the elastic modulus is 31.88–34.28 GPa and 28.43–31.47 GPa, respectively [13]. Kaiwen et al. indicate that laser energy input plays a significant role in determining formation qualities of the SLM-treated samples. High-density samples without obvious macro-defects can be obtained between 83 and 167 J/mm^3^ [21]. However, SLM involves complex thermophysical phenomena, such as laser scattering and absorption, rapid heat conduction, metal melting and solidification in a short time, and intense evaporation. Moreover, all of these physical phenomena occur in the melt pool with a size of hundreds of microns, making it very difficult to collect characteristic data of the SLM process, such as temperature revolution profile, velocity of the melt-pool flow, and melt-pool deformation history [22,23,24]. As a typical quantitative analysis method, numerical simulation has gradually become a powerful tool for studying the SLM process [25].

Due to the dominant position of laser heating in SLM processing, it is very important to consider the influence of thermo-fluid dynamics as fully as possible when modeling. Qiang Chen has divided the dominating thermal factors of SLM process into three categories, the laser as a heat source, evaporation that affects heat dissipation, and the violent flow in the melt pool caused by the Marangoni effect [26]. The interaction between laser and powder plays an important role for the reason that the laser is the driving force of the SLM process. Gusarov [27] described a model for assimilating powder into a uniform absorbing and scattering medium and found that the absorptance of a semi-infinite powder bed of opaque particles is a universal function of the absorptivity of the solid phase being independent of the specific surface and the porosity. Vaporization reduces the maximum temperature by taking away large amounts of energy, but it only occurs when the maximum temperature exceeds the boiling temperature [28]. The Marangoni effect is the tangential stress gradient caused by the temperature gradient on the surface of the molten pool. Dongyun Zhang [29] found that the Marangoni convection includes convective and conductive heat flux, and both of them have effects on molten pool shape, but the effect of convective heat flux is dominant. Trong-NhanLe [30] investigated the effects of Marangoni convection on the melt-pool formation during the selective laser melting of SS316 powder. The factors that promote the formation of the molten pool are summarized into three modes: the conduction mode, in which the melt-pool formation is dominated by thermal conduction; the keyhole mode, in which the melt-pool formation is determined mainly by the recoil pressure; and an additional transition mode that exists between these two modes, in which the melt-pool formation is driven mainly by the Marangoni convection effect. The author found that the Marangoni effect played a dominating role among the three models. In summary, the Marangoni effect is considered to have a significant influence on the flow generation in the melt pool [31].

At present, the numerical simulation research on the SLM process can be divided into two aspects. The first one is based on particle scale, focusing on investigating the complex reflection and absorption of laser between metal particles by building a numerical model [22,23,24,32,33]. The other is based on the scale of the part, treating the powder as a whole made of uniform material, and analyzing the temperature field and stress field during SLM process by numerical simulating [34,35]. However, due to the high thermal conductivity and low melting point of magnesium alloys, the current numerical simulation research on magnesium alloys processed by SLM is not very common. Mishra established a finite element model of SLM processing for AZ91D, studied the temperature change and the size of the melt pool during the processing, but did not conduct an in-depth study of the flow field in the melt pool [1]. The study of laser power on temperature distribution, melt-pool flow, and melt-pool morphology is not systematic enough. Therefore, it is of great importance and necessity to find a feasible method to reveal the relationship between the densification behavior and process parameters.

In this work, a numerical simulation model is presented, to investigate the thermophysical phenomena and their influence on the forming process of AZ91D alloy powder SLM, and the model is verified by experiments. Surface stress caused by the Marangoni effect and gravity in flow generation, the phase change latent heat, and Gaussian distributed laser beam were considered in building the model. The influence of temperature gradient and high cooling rate on the formability of final part is investigated and discussed. The convection flow in the melt pool and the resultant influence on melt pool dimensions are presented. The relation between melt-pool dimensions and shape with laser power is revealed.

## 2. Modeling

The Fluid Heat Transfer module and Laminar Flow module in COMSOL were involved in building the model. To simplify the model, the following assumptions are made in this study:(1)The powder material is considered to be a homogeneous whole, and the porosity of the powder material is indicated by mathematical methods.(2)The flow in the melt pool is considered to be incompressible laminar flow.(3)The evaporation of AZ91D magnesium alloy is ignored.

### 2.1. Physical Model

The X–Z plane is set as the symmetry plane in the model as shown in Figure 2. Thus, simulation can be performed on half of the model, in a symmetrical way, to reduce the calculation time. The calculation domain is divided into two layers. The upper layer represents the AZ91D powder material with a thickness of 0.04 mm, and the lower layer represents the formed part with a thickness of 0.12 mm. The size of the entire calculation domain is 0.2×0.6×0.16 mm. The laser heat source is defined as the commonly used Gaussian distribution surface heat source, perpendicularly incident from the upper surface of the model, and moving at a constant speed 1000 mm/s along the −X axis direction. At the same time, convective heat dissipation with nitrogen and heat radiation are also introduced on the upper surface of the model. The initial temperature of the domain is set as the room temperature. The parameters of the SLM process, such as laser power and scanning speed, are shown in Table 2, and the material properties of AZ91D are shown in Table 3.

### 2.2. Powder Bed Properties

The powder material is regarded as a whole in the numerical model, without considering the surface morphology and internal voids. Therefore, void parameters are introduced to characterize the effect of voids on thermal conductivity and density.
(1)$$\rho_{\mathrm{powder}}=\rho_{s}\left(1-\varphi \right)$$
(2)$$k_{\mathrm{powder}}=k_{s}\left(1-\varphi \right)$$
where $\rho_{\mathrm{powder}}$ is density of powder valued at 0.95 g/cm^3^ (see Section 3 for its testing). Thus, the value of φ can be calculated by Equation (1). The value of φ is 0.475.${\;k}_{\mathrm{powder}}$ is the dynamic viscosity of powder, and φ is the porosity of the powder.

### 2.3. Governing Transporting Equations

(1)Momentum conservation equation:

(3)$$\rho \frac{\partial \vec{u}}{\partial t}+\rho \left(\vec{\mu }\cdot 𝛻\right)\vec{u}=𝛻\cdot \left[-p\vec{I}+\vec{k}\right]+\vec{F}+\rho \vec{g}$$

(4)$$\rho 𝛻\cdot \left(\vec{u}\right)=0$$

(5)$$\vec{k}=\mu \left(𝛻\vec{u}+{\left(𝛻\vec{u}\right)}^{T}\right)$$

In Equation (3), ρ, $\vec{u}$, μ, and p represent density, velocity field, dynamic viscosity, and dynamic pressure, respectively. $\vec{I}$ is the three-dimensional unity tensor, and $\rho \vec{g}\;$ is the gravity force. $\vec{F}$ is a source term representing the sum of other body force, such as the buoyancy forces and the mushy-region flow resistance.

The Marangoni effect can be modeled as a shear stress on the upper surface of the melt pool, as in the following equation:(6)$$\left[-p\vec{I}+\mu \left(𝛻\vec{u}+{\left(𝛻\vec{u}\right)}^{T}\right)-\frac{2}{3}u\left(𝛻\cdot \vec{u}\right)\vec{I}\right]\vec{n}=\gamma {𝛻}_{t}T$$
where γ is surface tension.

(2)Energy conservation equation:

(7)$$\rho C_{p}\frac{\partial T}{\partial t}+\rho C_{p}\vec{u}\cdot 𝛻T+𝛻\cdot \vec{q}=Q+Q_{p}+Q_{vd}$$

(8)$$\vec{q}=-k𝛻T$$

Heat transfer in the process is described by Equations (5) and (6), in which the latent heat of phase change is also considered as follows:(9)$$\rho =\theta_{S}\rho_{S}+\theta_{L}\rho_{L}\;$$
(10)$$C_{P}=\frac{S}{\rho }\left(\theta_{S}\rho_{S}C_{P,S}+\theta_{L}\rho_{L}C_{P,L}\right)+L_{S\to L}\frac{\partial \alpha_{m}}{\partial t}$$
(11)$$\alpha_{m}=\frac{S}{L}\frac{\theta_{L}\rho_{L}-\theta_{S}\rho_{S}}{\theta_{S}\rho_{S}+\theta_{L}\rho_{L}}$$
(12)$$k=\theta_{S}k_{S}+\theta_{L}k_{L}$$
(13)$$\theta_{S}+\theta_{L}=S$$
where $\theta_{S}$ and $\theta_{L}$ are the mass ratio of solid phase and liquid phase, respectively.

(3)Heat source modeling

A Gaussian power distribution model is used as the laser heat source, which moves at a constant speed, v, along the −*X*-axis.
(14)$$Q_{laser}=\frac{2\varepsilon P}{\pi R^{2}}\exp\left(-\frac{2\left({\left(x-vt\right)}^{2}+y^{2}\right)}{R^{2}}\right)$$

$Q_{laser}$ is the input heat flux of the laser, ε is the emissivity of the powder, σ is Stefan–Boltzmann constant, *P* is the laser power, *R* is the Gaussian laser spot radius, v is the scanning speed, and *t* is the scanning time.

(4)Boundary conditions

The side and bottom surfaces are set as thermal insulation, and the Y–Z plane is set as a symmetrical surface to reduce the amount of calculation:(15)$$-\vec{n}\cdot \vec{q}=0$$

The boundary conditions on the top surface include laser beam deposition, forced convection heat dissipation, and heat radiation. The control equation is as follows:(16)$$-\vec{n}\cdot \vec{q}=-Q_{laser}+h\left(Text-T\right)+\varepsilon \sigma \left(Text^{4}-T^{4}\right)$$
(17)$$h=\{\begin{array}{c} 2\frac{k}{l}\frac{0.3387P_{r}^{1/3}R_{el}^{1/2}}{{\left(1+{\left(\frac{0.0468}{P_{r}}\right)}^{2/3}\right)}^{1/4}}\;ifR_{el}\leq 5\cdot 10^{5}\\ 2\frac{k}{l}P_{r}^{\frac{1}{3}}\left(0.37R_{el}^{\frac{4}{5}}-871\right)\;ifR_{el}>5\cdot 10^{5} \end{array}\;$$
where Text is the set ambient temperature valued at 293.15 K.

## 3. Results and Discussion

### 3.1. Temperature Distribution

Figure 3 shows the change of temperature field over time, when the scanning speed is v = 1000 mm/s and laser power is P = 100 W. The color map represents the temperature field, and the black line is the melt point isotherm of the AZ91D alloy which shows the boundary of the melt pool representatively. The temperature starts to rise as soon as the laser irradiates the upper surface of the powder, and the melt pool is generated immediately once the temperature exceeds the solidus temperature of the AZ91D alloy. With continuous laser irradiation, the volume of the melt pool increases rapidly and reaches a steady state after a short time. It has been found that the dimension of the melt pool stays almost constant after 300 μs (results are presented in the following section), which also shows that the SLM process has reached a stable state. Therefore, it seems that this model of 0.6×0.2×0.16 mm calculation domain can provide adequate information about the temperature distribution, flow in the melt pool, and melt-pool size.

Figure 4 shows the maximum temperature in the calculation domain with time when laser power is P = 100 W, which gives a sight of temperature change history in the domain. The maximum temperature rises rapidly to 1600 K in 10 μs, which contributes to a spectacularly high growth rate ($1.6\;\times 10^{8}\;K/s$ on average). However, the rise rate slows down, and the maximum temperature gradually stabilizes at around 1651 K after 300 μs. In order to reduce the amount of calculation, the computational domain should be as small as possible. By calculating with other conditions unchanged, the model with a scan path length of 1 mm found that the highest temperature of the computational domain gradually stabilized at 1651 K after 300 μs. Therefore, the currently selected computational domain size meets the simulation requirements. Since the fluctuation range of the maximum temperature does not exceed 1% after 300 μs, it can be considered that a stable state has been reached. This phenomenon in which the maximum temperature increases rapidly and reaches a steady state in a short time can be attributed to the high thermal conductivity of magnesium alloys. The high thermal conductivity of AZ91D leads to strong heat conduction, and, as a result, large amounts of heat are quickly transferred to the molded part and substrate.

The spatial temperature gradient is an important factor related to residual stress and microstructure of final parts. Therefore, it is important to study the temperature distribution along the scan direction and the temperature–time history of a significant point.

Figure 5 shows the temperature distribution along the scan path, at different laser power, when the laser incident center is at point A (0,0,0.16). It is obvious that there is an enhancement in maximum temperature with the increase of laser power. The ratio of the temperature drop to the corresponding distance is used to define the average temperature gradient. As laser power rises from 100 to 200 W, the maximum temperature rises from 1651 to 2739 K, as well. Due to high thermal conductivity of the AZ91D magnesium alloy, the temperature decreases along the scanning path at a very high speed, which reaches the highest value of $7.72\times 10^{6}$ K/m at P = 200 W. The great temperature gradient may lead to high stress field, and defects such as cracks that are caused by stress relief are likely to occur in final parts accordingly [37,38]. It is worth noting that the highest temperature point does not coincide with laser incident center, but slightly behind it, opposite to scan direction. This phenomenon is due to the combination of the thermal accumulation effect and the change of thermal conductivity caused by phase transition [39]. It can be seen from Figure 5 that the temperature curve is not axisymmetric, and the temperature drop rate is delayed between 0.12 and 0.18 mm. This results from the latent heat released during solidification of molten AZ91D alloy, when the temperature comes to solidus temperature.

The cooling rate is another significant factor determining the microstructure and mechanical properties of final parts. Figure 6 shows the temperature change curve of the fixed point B (0.1,0,0.16) on the scanning path, at different laser power levels. The cooling rate is presented by the average temperature drop rate of point B. It can be seen that the temperature of point B falls rapidly with a significantly high cooling rate, which varies from $4.31\times 10^{6}\;$ K/s at P = 100 W to $7.37\times 10^{6}\;$ K/s at P = 200 W. However, due to the latent heat of solidification, there is an obvious delay of the cooling rate when the temperature drops to the phase-transition temperature, as well. The cooling rate can affect the growth of magnesium alloy grains directly; a high cooling rate will refine solidified microstructure obviously. There are researches clarifying that a high cooling rate can make the eutectic distribute more homogeneously and its volume fraction decrease in Mg-Gd-Y-Zr alloy [40]. In addition, due to the high cooling rate of the melt pool during the SLM process, the formed material undergoes repeated remelting, and some elements in the powder material may be burned during the process, so the parts are prone to microsegregation and coarse crystals. The layer-by-layer melting and sintering of the powder bed may also lead to anisotropy of the mechanical properties of final parts [41,42,43].

Thus, compared with traditional manufacturing methods, SLM processing technology can produce parts with finer grains and better tensile yield strength, which is attributed to the high cooling rate during process. On the other hand, great temperature gradient due to high thermal conductivity can result in a high stress field, and defects such as cracks are likely to occur in the final parts. Therefore, extra high-input laser energy should be avoided in the SLM process, and substrate preheating can help to reduce thermal stress. Further research in this field requires experimental verification and will not be discussed in this article. However, the FE model can provide a deeper understand of the process and help to select suitable parameters with lower cost.

### 3.2. Flow in the Melt Pool

Thermocapillary flow (Marangoni convection) induced by surface tension variations along free surface is an important phenomenon in the SLM process [44], because it plays a crucial role in determining the dimension and stability of melt pool [41,42]. Figure 7 shows the shape of melt pool and flow in it at different laser power levels, when the melt-pool shape gets stable after 300 μs. The color map shows the temperature gradient, and the arrows represent the velocity field. It can be found that the dimensions of the melt pool, maximum temperature, and maximum speed of flow all increase with the enhancement of the laser power. For a relatively lower power of 100 and 125 W there is no remelting in the previously formed parts. Under this condition, the insufficient energy input may result in poor metallurgical bonding ability between the current processed layer and the previous processed one [38]. While at a considerably high laser power of 200 W, obvious remelting is observed in the previously formed layer, the maximum temperature is as high as 2738.53 K, and the maximum velocity in the melt pool is up to 10.14 m/s. In such a situation, overheating in the local area causes the capillary of the melt pool to be highly unstable, and the melt is liable to splash. Thus the spheroidization effect is likely to occur, which is a typical metallurgical defect in the SLM process. Thereby, the densification level of final part may be reduced [45]. Meanwhile, arrows show a radial flow on the upper surface of melt pool, diverging from the highest temperature in the center to the boundary of melt pool, which is consistent with the Marangoni effect.

Figure 8 shows the maximum velocity in the melt pool over time, at P = 100 W. The velocity in the melt pool increases rapidly in 150 μs, accompanied by the conspicuous rise of temperature. Moreover, the maximum speed reaches a basically stable state around 4.67 m/s, from 300 to 400 μs. The high-speed flow in the melt pool is caused by the tangential stress on the melt-pool surface which is contributed by the significant temperature gradient. Therefore, flow rate is positively correlated with temperature. Maximum velocity increases and stabilizes, accompanied with maximum temperature, as can be seen from Figure 3 and Figure 8.

The maximum velocity in the melt pool varies from 4.67 m/s at P = 100 W to 10.14 m/s at P = 200 W. Such violent flow will greatly enhance the convective heat transfer in SLM, making it significant to study the flow in the melt pool. Figure 9 shows the velocity vector of the convective flow in the melt pool, at P = 100 W. The color map represents different speeds, and the arrows represent the flow direction. The maximum flow velocity occurs between the laser incident center and the melting front of the melt pool, which is accompanied by the most significant temperature gradient. A high temperature gradient is the root cause of flow. A great surface tension gradient is caused by the temperature gradient; furthermore, the generated tangential force pushes melted magnesium alloy to flow at high speed (the maximum velocity reaches 4.67 m/s on the upper surface).

Molten alloy flows from the center to the edge on the upper surface of the melt pool (Figure 9a), and then it flows from the upper surface to the bottom, along the boundary of the melt pool (Figure 9b). The flow converges at the center of the melt pool bottom and rises from the bottom to the upper surface (Figure 9c). In general, the shape of the melt pool is determined by two forms of heat transfer in the melt pool: conductive heat transfer and convective heat flux. Moreover, the latter plays a decisive role in SLM [29,46]. The radial flow takes a large amount of heat from the laser incident center to the melting boundary and solidification boundary, which promotes the boundary expansion. The width and length of the melt pool are expended as a result. Meanwhile, high-speed flow on the melt-pool surface can greatly accelerate the heat loss to atmosphere, resulting in a reduction of heat conducted to the bottom of the melt pool, contributing to the decrease of melt-pool depth. The melt pool becomes larger but shallower in the end. Thus, metallurgical bonding between the current processed layer and previous processed one may be weakened, and porosity and other defects are more likely to occur.

Since the model does not take evaporation into account, which can greatly promote heat conduction of the melt pool, the flow velocity should be slower, and the shape of the melt pool will be narrower and deeper when it comes to the experiment. Nevertheless, the simulation model still can help to see the flow phenomenon in the melt pool. Moreover, the influence of the high-speed flow on heat dissipation and evolution of melt-pool morphology can be investigated by the FE model.

### 3.3. Melt-Pool Size and Shape

In the current SLM process simulation, the size of the melt pool was mostly determined on the basis that the temperature is higher than the melting point of the powder material [47]. Powder begins to melt when the temperature exceeds the solidus temperature (743 K), and it completely melts when the temperature exceeds liquidus temperature (868 K), at which the semi-melted parts exist between them will adhere to the final part. Therefore, the actual boundary of melt pool will be beyond the boundary of the melting point. In order to verify the model through experiments better, a phase-transition model is introduced. Governing equations of phase transition are Equations (9)–(11) in the former section, and ω is introduced in to describe the state of the AZ91D alloy as shown in Table 4.

Different colors in Figure 10 indicate different states of powder: The red region indicates that the powder is totally melted (where ω = 1), the blue region represents the part that has not begun to melt (ω = 0), and the color-transition zone shows the semi-melted part. The simulation was conducted with laser power at P = 100 W.

Figure 11 shows the melt-pool size with time at P = 100 W. It can be seen that the size of the melt pool (length, width, and depth) increases rapidly with time in the initial period, and the growth rate slows down after 50 μs, until it reaches a steady state. The depth of melt pool stabilizes at 40 μm, first, and then the width stabilizes at 171.4 μm. The length finally reaches a stable value of 220 μm after 300 μs. This is because the residual temperature and latent heat of solidus reduce heat dissipation in the rear end of the melt pool along the scan direction.

Figure 12 shows the different melt-pool dimensions at different laser power levels. It is obvious that the three dimensions (length, width, and depth) all increase with the laser power, as a result of the increased input energy of the laser, as discussed in the former section. However, it is worth noting that the length of the melt pool increases faster with the increase of laser power, compared to the width and depth. The tail of the melt pool is surrounded by a compact formed part with high thermal conductivity, while the melting front and two sides of the melt pool are adjacent to the unmelted powder. With the increase of laser power, large amounts of heat are transferred to the solidification front of the melt pool through the formed part with higher thermal conductivity, resulting in a higher ascent speed in the length of the melt pool.

Increased laser power can also affect the shape of the melt pool. The variable w/l (width/length ratio) is introduced, to characterize the shape of the molten pool along the scanning direction, while the variable d/w (depth/width ratio) is used to describe the shape in the depth direction. Figure 13 shows the two variables changing trends with different laser power levels. It can be seen that the two variables develop in opposite trends as the laser power increases. When the laser power is less than 175 W, w/l continues to decrease. However, it rises slightly when the laser power comes to 200 W. On the contrary, d/l continues to increase with the increase of laser power, but the rate of increase gradually drops down, and it is almost stable when the laser power increases from 175 to 200 W.

In summary, the melt pool became narrow and deep as the laser power increased, which can lead to poor stability of the melt-pool shape. Coupled with the violent evaporation caused by high laser power, it is easy to cause defects such as pores in the final part. On the other hand, it has been analyzed before that lower laser power will lead to an insufficient melt-pool size, which can result in inadequate bonding quantity between the forming layers. Therefore, laser power of 100 W and layer thickness of 0.04 mm are the most reasonable processing parameters for such an experiment.

### 3.4. Experiment Verification

The width of the laser track, which is strongly related to the scanning patch, is an important factor affecting the final material. Therefore, it is considered to be an effective verification method to compare the simulation width with experiments [48]. In order to verify the credibility of this model, two different experiments were conducted to observe the width of the melt channel. Then the melt-pool width in the simulation model was compared with the width measured in experiments.

Gas atomized AZ91D magnesium alloy powder (Supplier: Tangshan Weihao Magnesium Powder Co., Ltd. Tangshan, China) with a mean particle size of 53 μm was used in the experiments. An Nd:YAG laser with 1064 nm wave length and 80 μm spot size was used to scan and melt the powder. The process was protected by argon, and the oxygen concentration in the building chamber was below 0.1%.

Experiment I is a multi-track scheme. Square areas 10 mm × 10 mm were melted in the beds’ surface, with laser power 100 W and scanning speed 1000 mm/s. The laser scanned parallel to a side of the square area, with scan spacing of 0.2 mm, and the thickness of each layer was 0.04 mm. A typical result of this experiment is shown in Figure 14. Twenty data of melt channel width were collected from the surface of final part by AXIO Scope A1 of Carl Zeiss (Produced by Carl Zeiss, Germany). The average is considered to be the experimental width of the single melt channel. The experimental width result is 153.43 μm, as shown in Table 5, which is slightly lower than the simulation of the melt channel width of 171.40 μm, with deviation of 10.65%.

The evaporation temperature of the AZ91D alloy was 1373.15 K, while the maximum temperature of the molten pool was 1660 K in the simulation, which is much higher than the evaporation temperature. Therefore, violent evaporation will occur on the surface of the melt pool, which can accelerate the loss of heat into the gas environment. The heat transferred to the powder material is reduced as a result, so the width of the melt pool in the experiment is smaller than the result obtained by simulation.

Experiment II was carried out in a more intuitive way. A straight line of 5 mm was melted with laser power 100 W and scanning speed 1000 mm/s. The laser scanned six layers of AZ91D alloy powder, and the thickness of each layer was 0.04 mm. The experimental results are shown in Figure 15. Twenty data are collected from the surface of final part by AXIO Scope A1 of Carl Zeiss. The average is considered as the experimental width of the single melt channel. The average width of the single melt channel in this experiment is 173.95 μm, which is slightly larger than the simulation result of 171.40 μm with an error of 1.49%. The results are listed in Table 6.

The result of experiment II is closer to the FE model. However, because the semi-melted powder particles adhered to the melt pool (reflective particles in Figure 15), the experimental results are slightly larger than the simulation results. However, the experiment results are in good agreement with the simulation results, with an error of 1.49%, which proves the reliability of the model directly.

## 4. Conclusions

A three-dimensional model of selective laser melting of AZ91D magnesium alloy was established, in which the phase change latent heat and Marangoni floe was considered. The model was used to investigate the temperature profile of the domain, flow behavior of the melt pool and melt-pool dimensions and shape under different laser power. The model can help to investigate the complex physical phenomenon in SLM and select proper processing parameters with lower cost. Conclusions can be drawn as follows:(1)The temperature of the domain increased rapidly when the laser started to irradiate, and the maximum temperature showed a strong correlation with input laser energy. Severe temperature gradient and high cooling rate were observed, which is attributed to the high thermal conductivity of AZ91D magnesium alloy.(2)Violent flow caused by the Marangoni effect was observed in the melt pool, and the maximum velocity was found to increase when the laser power would rise. The high-speed flow enhanced convection heat transfer and made the melt pool wider and longer but shallower.(3)The dimensions of the melt pool stabilized rapidly., accordingly with the temperature and flow velocity. The size of the melt pool increased as the laser power rose, but the length of the melt pool increased faster than its width and depth. The shape of the melt pool became narrower and deeper as the laser power increased, which may lead to poor stability of the melt pool’s shape.(4)The width of the melt pool was acquired by single scan melt experiment, which was in good agreement with the results predicted by the FE model (with an average error of 1.49%).

## Figures and Tables

**Figure 1 materials-13-04157-f001:**
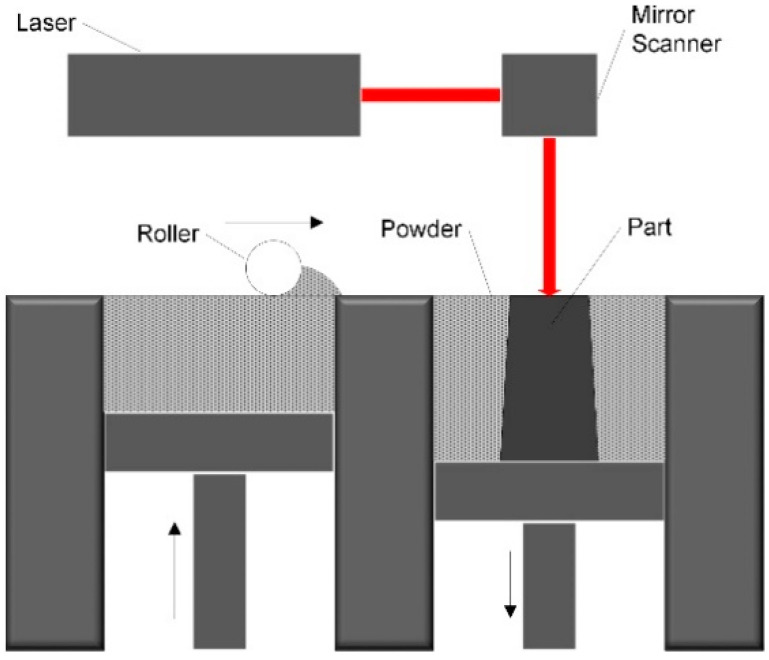
Schematic of selective laser melting (SLM) process.

**Figure 2 materials-13-04157-f002:**
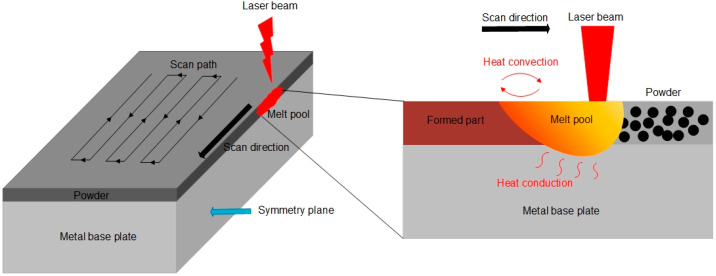
Single scan model geometry.

**Figure 3 materials-13-04157-f003:**
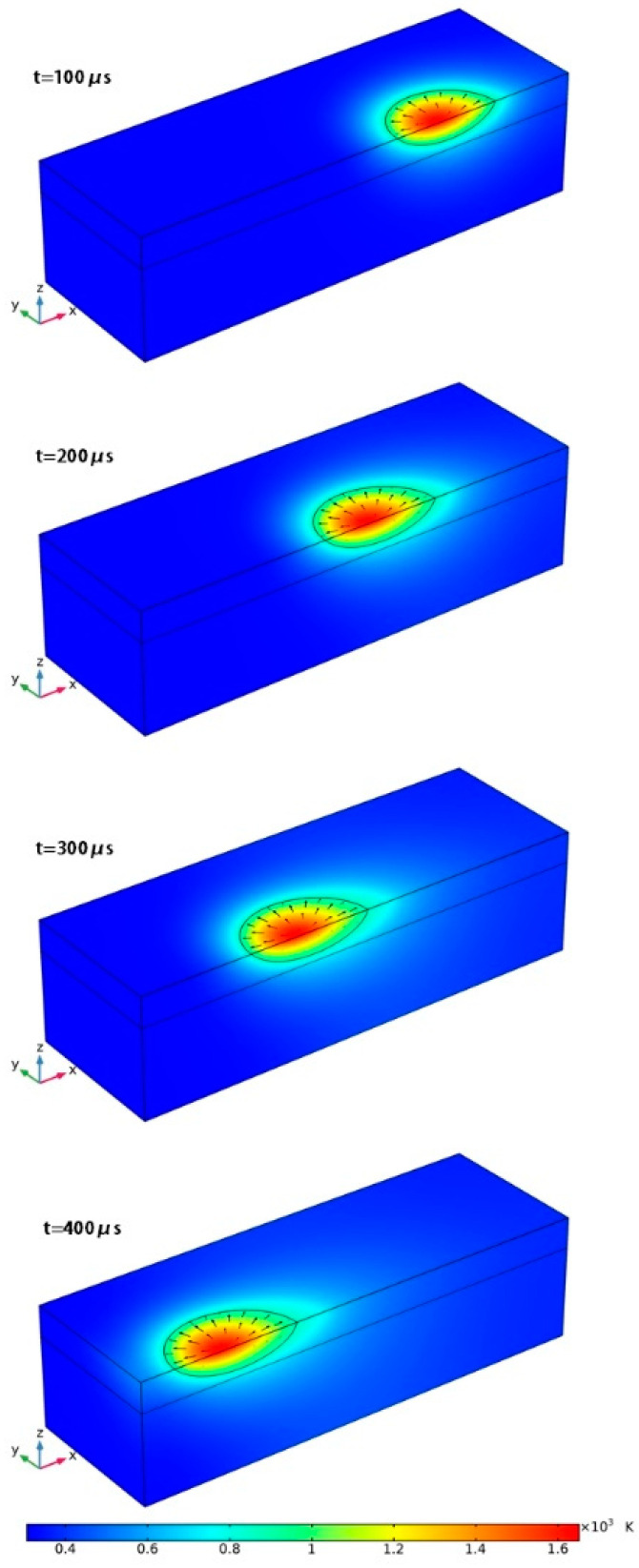
Temperature evolution with time at P = 100 W.

**Figure 4 materials-13-04157-f004:**
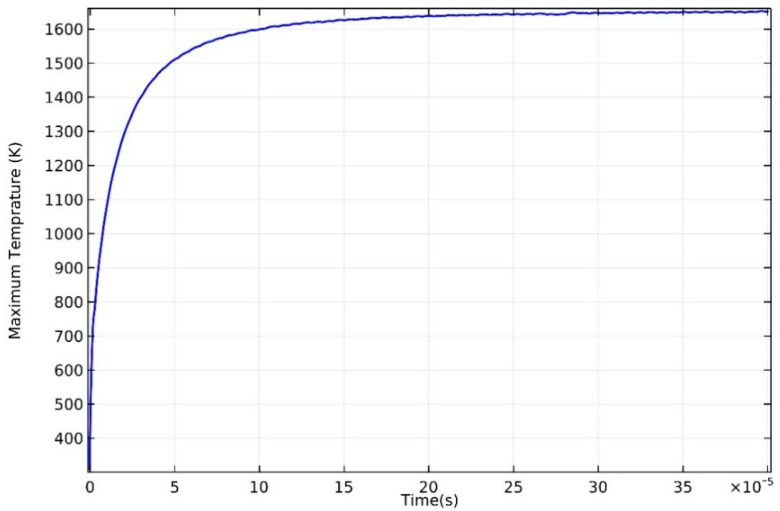
Maximum temperature of domain vs. time at 100 W.

**Figure 5 materials-13-04157-f005:**
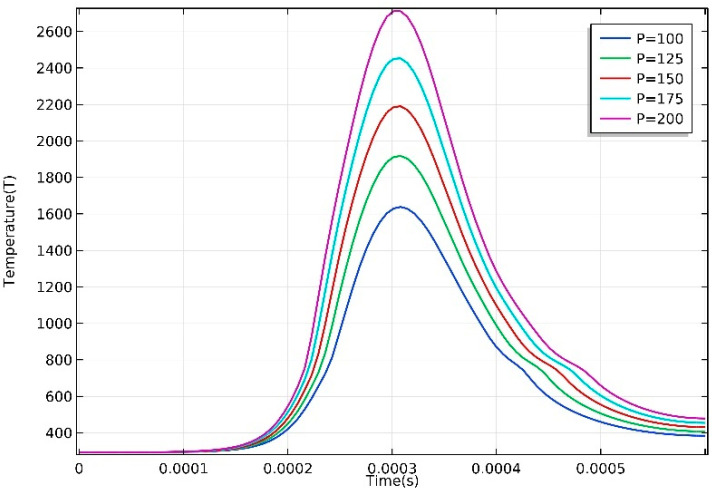
Temperature along the scan path.

**Figure 6 materials-13-04157-f006:**
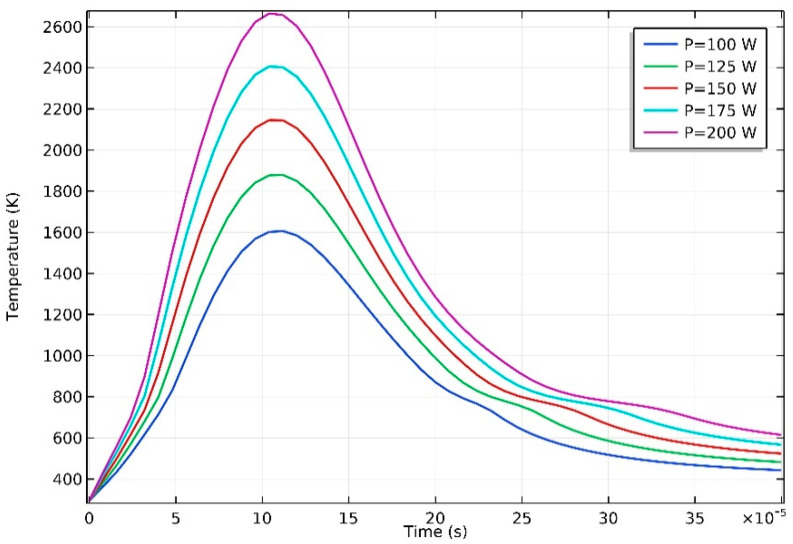
Temperature of point B with time.

**Figure 7 materials-13-04157-f007:**
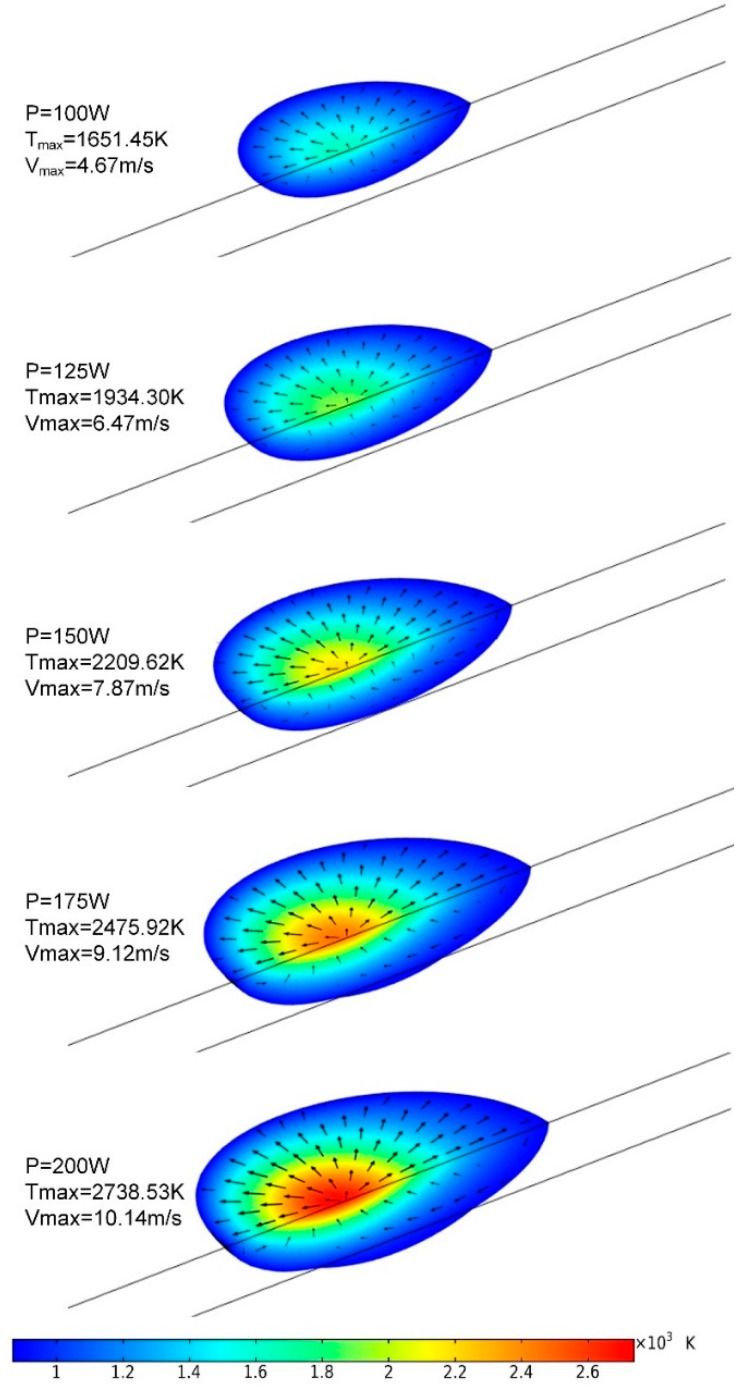
Melt-pool comparison for different laser power levels (*t* = 300 μs).

**Figure 8 materials-13-04157-f008:**
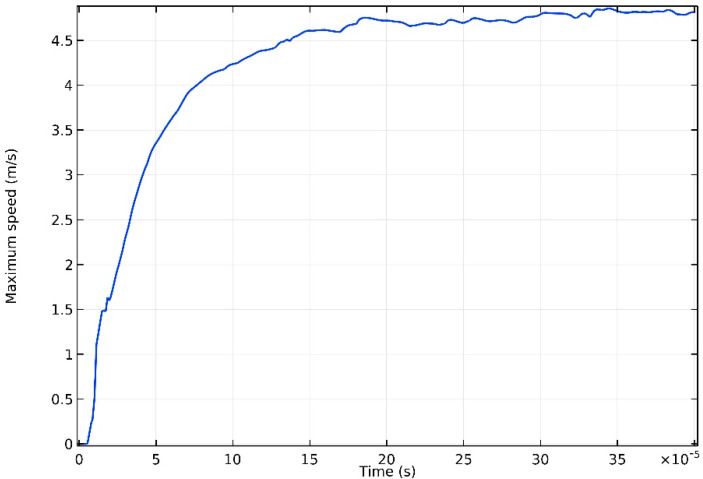
Maximum speed in the molten poll, at P = 100 W.

**Figure 9 materials-13-04157-f009:**
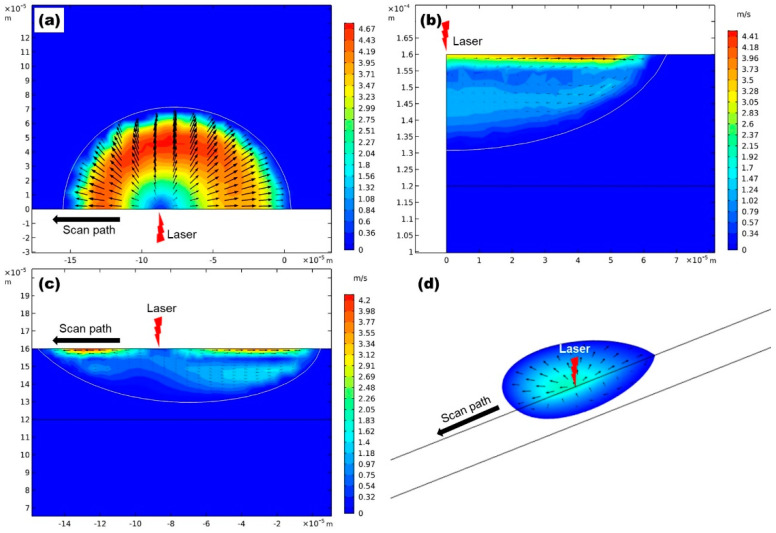
The velocity vector plots of convective flow in the melt pool, at P = 100 W. (**a**) shows the flow on the upper surface of melt pool, (**b**) shows the flow on the Y-Z section through the laser incident center, (**c**) shows the flow on the X-Z section trough the laser incident center. (**d**) shows the shape of the melt pool and the flow in it at P = 100 W and t = 300 μs.

**Figure 10 materials-13-04157-f010:**
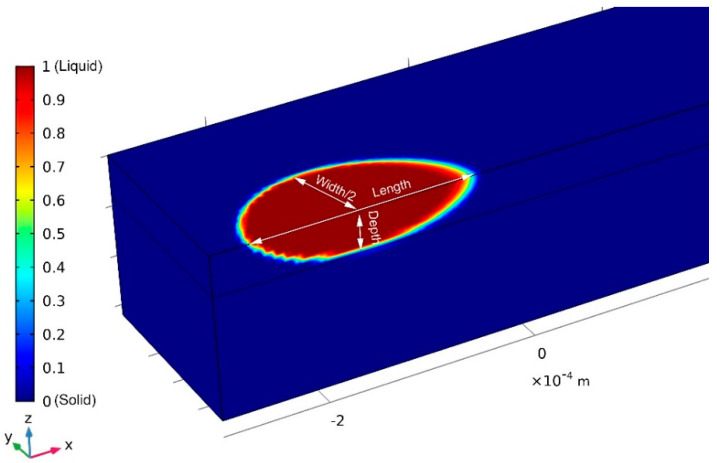
Schematic diagram of melt-pool size at 400 µs and laser power at P = 100 W.

**Figure 11 materials-13-04157-f011:**
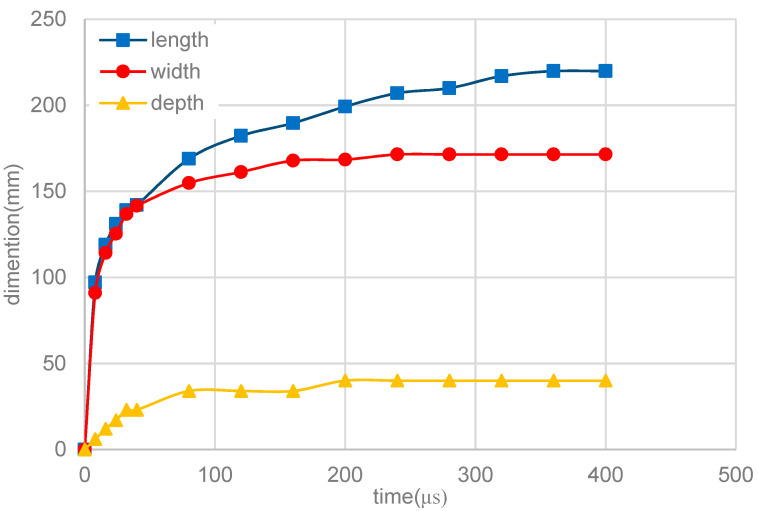
Melt-pool size with time at P = 100 W.

**Figure 12 materials-13-04157-f012:**
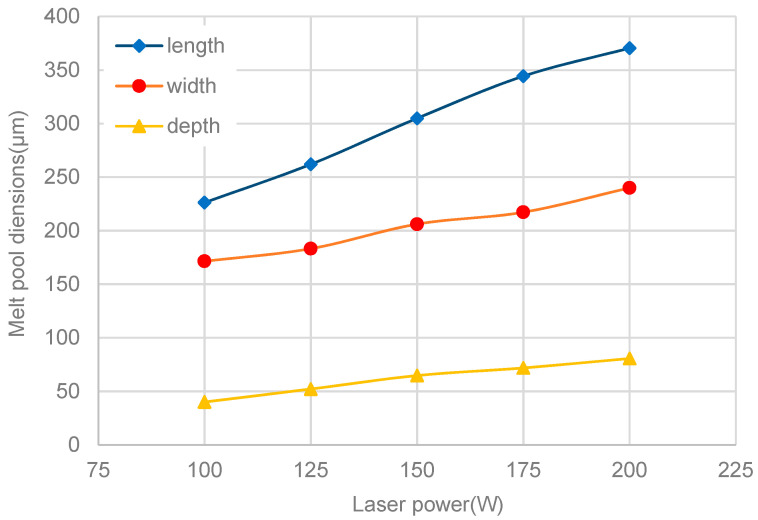
Melt-pool dimensions at different laser power levels.

**Figure 13 materials-13-04157-f013:**
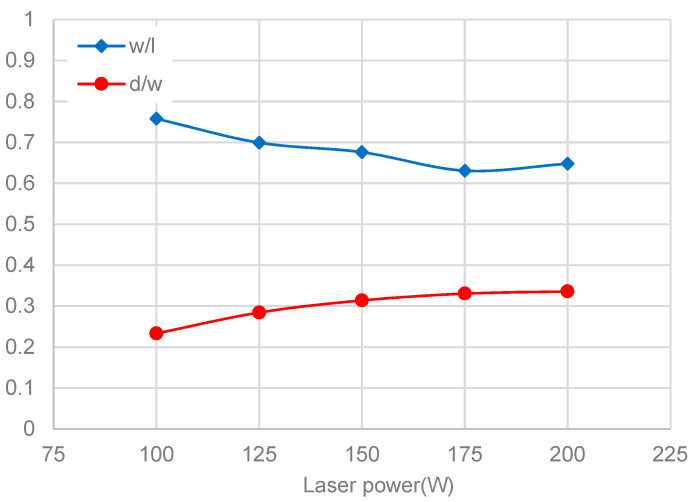
Ratio of width/length and depth/width at different laser power levels.

**Figure 14 materials-13-04157-f014:**
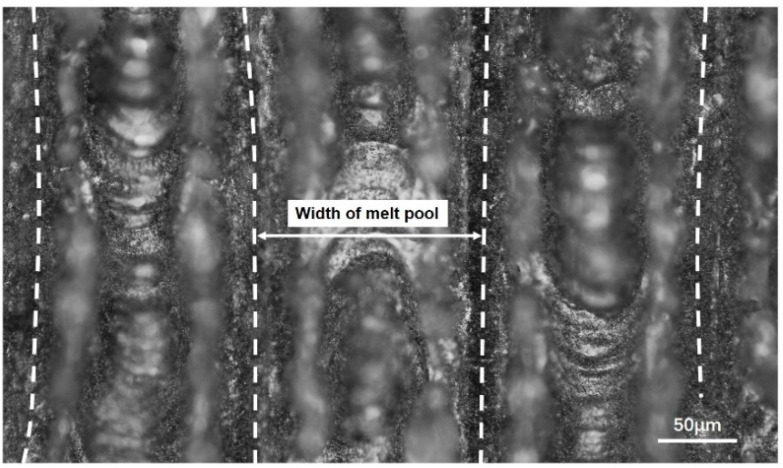
Multi-channel scan experiment at P = 100 W.

**Figure 15 materials-13-04157-f015:**
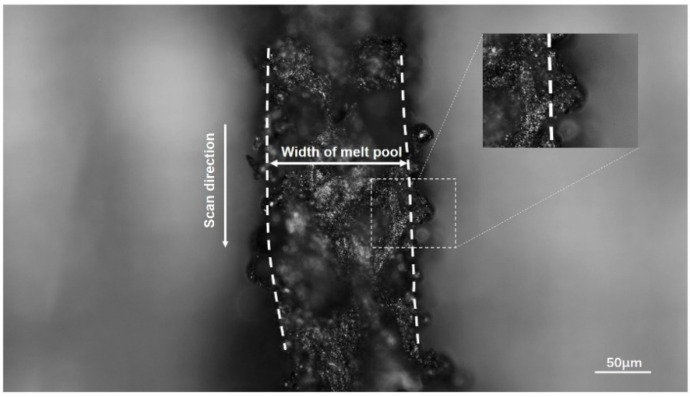
Single-channel scan experiment at P = 100 W.

**Table 1 materials-13-04157-t001:** Compositions of impurities in AZ91D magnesium alloy.

Chemical Composition	Al	Zn	Mn	Fe, Cu, etc.	Mg
Content (wt%)	9.08	0.65	0.23	≤0.0051	others

**Table 2 materials-13-04157-t002:** Process parameters for simulation.

Computational domain dimensions (mm)	0.6×0.2×0.16
Layer thickness (mm)	0.04
Laser spot size (mm)	0.08
Laser power (W)	100, 125, 150, 175, 200
Scan velocity (mm/s)	1000
Porosity (φ)	0.475

**Table 3 materials-13-04157-t003:** Properties of AZ91D alloy (Friedrich and Mordike, 2006) [36].

Solidus temperature $\left(T_{S}\right)$	743 K
Liquidus temperature $\left(T_{L}\right)$	868 K
Specific heat capacity $\left(C_{P,S}\right)$	1014 J/kg-K (solid, 293 K)
$C_{P,L}$	1230 J/kg-K (liquid)
Thermal conductivity $k_{S}$ (Solid)	72 × 10^3^ W/K
$k_{L}$(Liquid)	82.9 × 10^3^ W/K
Latent heat of fusion L	373 kJ/kg
Dynamic viscosity µ	3 × 10^−3^ Pa∙s
Thermal expansion coefficient	2.6 × 10^−5^ K^−1^
Volumetric thermal expansion coefficient ($\beta_{T}$)	9.541 × 10^−5^ K^−1^
Temperature coefficient for surface tension (𝛛γ/𝛛T)	−2.13 × 10^−4^ N/m-K
Emissivity (ε)	0.18

**Table 4 materials-13-04157-t004:** State of AZ91D magnesium powder represented by ω.

ω	State of Powder
ω=1	Liquid (completely melted)
0<ω<1	Semi-melted
ω=0 **.**	Solid (unmelted)

**Table 5 materials-13-04157-t005:** Comparison between average width in experiment I and the model melt-pool width.

Average width of melt channel in experiment I (μm)	153.14
Average width of melt pool in simulation model (μm)	171.40
Error	10.65%

**Table 6 materials-13-04157-t006:** Comparison between average width in experiment II and the model melt-pool width.

Average width of melt channel in experiment II (μm)	173.95
Average width of melt pool in simulation model (μm)	171.40
Error	1.49%

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
