# Peer review of "Thermo-Fluid-Dynamic Modeling of the Melt Pool during Selective Laser Melting for AZ91D Magnesium Alloy"

_materials, 2020, doi:10.3390/ma13184157_

Round 1

Reviewer 1 Report

Please refer to attached review comments

Reviewer 2 Report

Dear authors,

the presented study of modelling and experimental validation of the melt pool of an AZ91 D alloy is designed well and also conducted in a thorough way.

The introduction gives a good overview of manufacturing Mg alloys, the application of AM and also the problems related to AM.

The language of the introduction is better than in the other sections of the manuscript, that has to be re-edited.

The modeling (the physical background) and the parameter selection seem reasonable to me and are described well. The experiment design and the results are plausible.

in section 2.3 there is an i as superscript ( ... effecths when this gradient is due to temperature difference. i) but I cannot find any text associated with that.

Fig. 5: please indicate the laser position

Concerninr rapid solidification and frequent re-heating: precipitation of primary and secondary phases should also be considered or at least mentioned.

Fig 10: although the colors are described in the text, I would recommed to indicate the areas in the scale bar

Section 3.3: please use evaporation instead of gasification

Fig. 13: Please add the directions of incident laser, direction of movement, layers, etc to the figure

In general the influences on the melt pool are described well as well as the possible consequences of parameter changes on the final material.

I strongly recommend professional english editing or proof-reading by a native speaker as there are many sentences with missing prepositions, incorrect word order  and inappropriate tenses.

However, the changes I recommend are of minor type and should not hinder publication of the manuscript

Reviewer 3 Report

Dear Editor: I would like to express my deep thanks for inviting me to review the manuscript ID: materials-899402

Title:       Thermo-fluid-dynamic Modeling of the Melt Pool during Selective Laser Melting for AZ91D  

Authors: Hongyao Shen, Jinwen Yan, Xiaomiao Niu

Comments:

Title:

Thermo-fluid-dynamic Modeling of the Melt Pool during Selective Laser Melting for AZ91D

Replaced by

Thermo-fluid-dynamic Modeling of the Melt Pool during Selective Laser Melting for AZ91D magnesium alloy

Abstract:

Please rewrite the abstract according to your results.

Please mention AZ91D composition

Introduction part:

Please write the aim and novelty in this work at the end of introduction section.

 “steel (2/9th of density at room temperature). [1]”

replaced by throughout the manuscript

 “steel (2/9th of density at room temperature) [1].”

Results and discussion:

  1. Why you choose only 100w and 200w instead of others.
  2. In figure 11 please add P=200W data.
  3. Figure 12 and 13 is not clear. Please provide clear images.
  4. According to Figure 14 experimental results differ significantly as compare to the simulation results please clarify this point.
  5. Please give a detail discussion of all Figures.

Conclusion part:

Please rewrite the conclusion according to experimental and simulation results.

RECOMMENDATION

After reviewing the enclosed manuscript for “Materials”, the present manuscript contains some kinds of scientific analysis but it is mandatory required to modify according to the preceding remarks. So, the manuscript can be accepted for publication after major mandatory revisions have been made.

Round 2

Reviewer 1 Report

I have quickly browsed through the revised manuscript and continue to strongly
feel that the content of the manuscript is based on several discrepancies and
scientifically flawed principles hence I recommend that the article should not
be considered for publications.

Reviewer 3 Report

Author addressed all of my comments in revised manuscript.